# If an LLM Were a Character, Would It Know Its Own Story?
# Evaluating Lifelong Learning in LLMs

## Abstract

Large language models (LLMs) can carry out human-like dialogue, but unlike humans, they are stateless due to the superposition property. However, during multi-turn, multi-agent interactions, LLMs begin to exhibit consistent, character-like behaviors—hinting at a form of emergent lifelong learning. Despite this, existing benchmarks often fail to capture these dynamics, primarily focusing on static, open-ended evaluations. To address this gap, we introduce LifeState-Bench, a benchmark designed to assess lifelong learning in LLMs. It features two episodic datasets—Hamlet and a synthetic script collection—rich in narrative structure and character interactions. Our fact-checking evaluation probes models' self-awareness, episodic memory retrieval, and relationship tracking, across both parametric and non-parametric approaches. Experiments on models like Llama3.1-8B, GPT-4-turbo, and DeepSeek R1, we demonstrate that non-parametric methods significantly outperform parametric ones in managing stateful learning. However, all models exhibit challenges with catastrophic forgetting as interactions extend, highlighting the need for further advancements in lifelong learning.

## 1 Introduction

Large language model (LLM)-based dialog agents exhibit human-like traits (*e.g.,* intent understanding and language expression), making users prone to anthropomorphism Shanahan et al. (2023). However, LLMs differ from humans in their *superposition property* Janus (2022): initially existing as a stateless superposition of simulacra across multiple possible characters Lu et al. (2024). This property emerges from its next-token prediction training on a massive corpus, whereas humans develop through accumulated experiences and memories.

Through sustained interaction, we observe that an initially **stateless** LLM can transition toward more **stateful** characteristics as dialogue context accumulates. At first, an LLM holds multiple characters but gradually settles into a clear character as the dialogue continues. Taking a nuanced view,

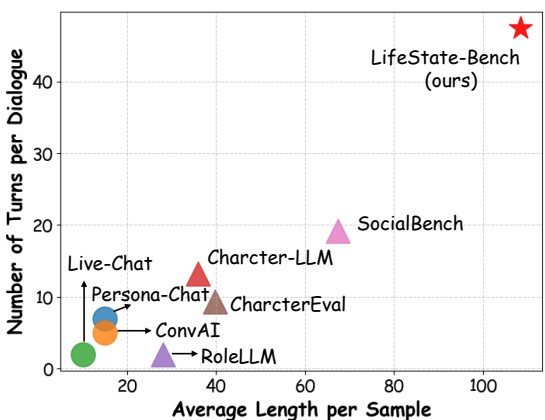

Figure 1: Dataset Statistics. Triangles represent role ability benchmarks, while circles denote dialogue agent benchmarks.

this character convergence process mirrors how humans update their state through accumulated experience.

This state transition raises a measurable question: How can we quantify an LLM's state evolution (also called Lifelong learning ability) from superposition to a more consistent state during multi-turn, multi-agent interactions? In this paper, "state" refers to the evolving configuration of an LLM's

internal processes during multi-agent interactions Adams et al. (2012); Sumers et al. (2024), building on AI cognitive architecture Sun (2004); Newell (1980).

While this research question predates LLM area, current exploration remains preliminary with varying methodologies. Early Persona-Chat series Gao et al. (2023); Zhang et al. (2018); Dinan et al. (2019) focusing on consistent character responses using seq2seq models, or design social intelligence questionnaire-based benchmarks Sap et al. (2019); Le et al. (2019). Both limited by static, non-interactive setups. Ground truths were either open-ended or fixed over time.

Generative agents Park et al. (2023) bring LLM-based dialogue agents into interactive human behavior simulation. This opens new possibilities for modeling state transitions. Later works follow two directions. First, role ability benchmarks Tu et al. (2024); Wang et al. (2024a); Shao et al. (2023) focus on role-playing and plot prediction. They improve dialogue realism, but place less emphasis on tracking factual states during interactions. Second, the Sotopia series Zhou et al. (2024); Wang et al. (2024b) and SocialBench Chen et al. (2024) accessing social intelligence in open-ended tasks. Their design often centers around user-defined social goals, which may not align with factual state tracking or verification.

To address these challenges, we propose LIFESTATE-BENCH to explore and measure LLMs' lifelong learning capabilities. As shown in Figure 1, our benchmark surpasses others (*e.g.,* dialogue agents, role-playing) with longer average sample lengths and more dialogue turns per interaction. Key features include:

**Cumulative Experience.** Inspired by the idea that "human personality emerges from experiences" Shao et al. (2023), we created an episodic dataset with clear timelines. Each episode contains scene details, character actions, and dialogues to enable continuous agent interaction.

**Fact Checking.** Each episode includes fact-based questions related to self-awareness, memory retrieval, and relationship changes, accompanied by reference answers to ensure objective evaluation.

**Memory Testing.** For lifelong learning evaluation, models should retain long-term memory of past scenes while accessing only recent dialogue. This is tested via (i) non-training methods: episode or summary concatenation, and (ii) training methods: knowledge editing Wang et al. (2025); Meng et al. (2023) and LoRA fine-tuning Hu et al. (2022) using historical context.

In LIFESTATE-BENCH, we selected theatrical scripts, including both existing (*e.g.,*, Hamlet) and synthetic narratives. For existing works, such as Hamlet—a classic play likely present in pretraining corpora—we use them to assess the model's memory retention capabilities. To reduce direct string matching, all character names have been anonymized. In contrast, the synthetic scripts, generated by Claude and unseen during pretraining, are used to evaluate the model's ability to adapt to entirely new content. This contrast allows us to **explore lifelong learning in a realistic setting**, where models must navigate both familiar and novel domains. Compared to current benchmarks, our dataset features more interactive characters, closed dialogue turns, and richer content (Table 1). Evaluation combines LLM-as-judge with human assistance, using predetermined factual answers as criteria.

We tested several popular models, including the open-source Llama3.1-8B AI (2024), the closed-source GPT-4-turbo OpenAI (2023), and the large language reasoning model DeepSeek R1 DeepSeek-AI et al. (2025). Benchmark-backed experiments show that current models still have much room for improvement in lifelong learning.

In summary, our work contributes in three key areas:

• **Two Datasets:** We introduce the Hamlet and synthetic datasets, featuring multi-agent episodic timelines and scene details to simulate cumulative experiences.

• **A Benchmark:** LIFESTATE-BENCH evaluates LLMs' lifelong learning abilities via fact-checking mechanism, using both non-parametric and parametric memory-testing methods.

• **Findings and Implications:** Non-parametric methods outperform parametric ones in lifelong learning, but all models still face challenges with catastrophic forgetting as episodes progress, suggesting that our benchmark could provide valuable insights for further improvements.

| Benchmarks | Dataset Characteristics | | | | | Interaction Design | | | Evaluation Focus | |
| --- | --- | --- | --- | --- | --- | --- | --- | --- | --- | --- |
| | # Samples | Avg Length | Data Source | # Turns | # Agents | Query Type | Answer Type | State | Memory | Metrics |
| *Dialog Agent Benchmarks* | | | | | | | | | | |
| PERSONA-CHAT Zhang et al. (2018) | 162.0K | 15 | Crowd | 7 | 2 | Chit-chat | Open | ✓ | ✓ | PPL, F1, Hit@1 |
| ConvAI Dinan et al. (2019) | 131.0K | 15 | Crowd | 5 | 2 | Chit-chat | Open | ✓ | ✓ | PPL, F1, Hit@1 |
| Live-Chat Gao et al. (2023) | 9.4M | 10 | Crawled | 2 | 2 | Chit-chat | Open | ✗ | ✗ | BLEU, ROUGE |
| MT-Bench Zheng et al. (2023) | 3.3K | 373 | Synthetic | 2.9 | 2 | Multi-task | Factual | ✗ | ✗ | Model Judge |
| *Role Ability Benchmarks* | | | | | | | | | | |
| Character-LLM Shao et al. (2023) | 21.1K | 36 | Synthetic | 13.2 | 2 | Persona | Open | ✓ | ✗ | Model Judge |
| RoleLLM Wang et al. (2024a) | 168.1K | 28.1 | Crawled | 2 | 2 | Persona | Mixed | ✗ | ✗ | ROUGE, Model Judge |
| CharacterEval Tu et al. (2024) | 11.4K | 39.8 | Crawled | 9.3 | 2 | Persona | Open | ✗ | ✗ | Model Judge |
| SocialBench Chen et al. (2024) | 30.8K | 67.4 | Synthetic | 19.2 | 3.8 | Social | Mixed | ✗ | ✓ | Model Judge |
| *Long-context Understanding Benchmarks* | | | | | | | | | | |
| Long Range Arena Tay et al. (2021) | - | 10.0K | Synthetic | 1 | 1 | Multi-modal | Factual | ✗ | ✗ | Acc, Speed |
| LongBench Bai et al. (2024) | 4.6K | 10.0K | Synthetic | 1 | 1 | Multi-task | Factual | ✗ | ✗ | Acc, F1, ROUGE |
| L-Eval An et al. (2024) | 411 | 4K-60K | Synthetic | 1 | 1 | Multi-task | Mixed | ✗ | ✗ | ROUGE, Model Judge |
| ∞-bench Zhang et al. (2024) | 130 | 200.0K | Synthetic | 1 | 1 | Multi-task | Factual | ✗ | ✗ | Model Judge |
| LIFESTATE-BENCH-Hamlet | 1.3K | 125.5 | Crawled | 66.1 | 6.6 | Social+Memory | Factual | ✓ | ✓ | Model Judge |
| LIFESTATE-BENCH-Synth | 202 | 91.9 | Synthetic | 28.9 | 7 | Social+Memory | Factual | ✓ | ✓ | Model Judge |

Table 1: Comparison of Different Benchmarks. ✗: not supported; ✓: fully supported. Data Source indicates the origin of the data. # Turns shows the average conversation turns. # Agents indicates the number of participants in each interaction. Query Type shows the question/task type. Answer Type indicates whether the expected answers are open-ended, factual, or mixed. State shows whether the benchmark maintains interaction state. Memory indicates whether the benchmark evaluates memory capability.

## 2 RELATED WORK

**Anthropomorphic Cognition in LLMs.** Early cognitive science Sumers et al. (2024); Laird et al. (1987); Sun (2004) laid the foundation for anthropomorphizing AI, simulating human-like emotional and social behaviors. Role-playing language agents have become increasingly common in simulating collective social behaviors in multi-agent systems. These agents Park et al. (2023) not only enhance social interactions but also contribute to personalized and complex task execution in AI.

**Role Ability/Dialog Agents Benchmarks.** Role ability Shao et al. (2023); Wang et al. (2024a) and dialogue agent benchmarks Zhang et al. (2018); Dinan et al. (2019); Gao et al. (2023); Zheng et al. (2023) are divided into static and dynamic types. Static models Chen et al. (2023); Tu et al. (2024) focus on predefined roles and fixed interaction patterns, typically applied in basic dialogue tasks. In contrast, dynamic models Chen et al. (2024); Zhou et al. (2024); Wang et al. (2024b) allow agents to accumulate experiences and evolve during interactions, enabling consistency and adaptability over time. These benchmarks are essential for evaluating agent flexibility, memory handling, and long-term interaction capabilities.

**Long-context Understanding Benchmarks.** Long-context understanding involves models processing large amounts of information over extended interactions. These benchmark Tay et al. (2021); Bai et al. (2024); An et al. (2024); Zhang et al. (2024) tests an agent's ability to synthesize and recall information from multiple episodes, maintaining coherence across long spans of dialogue. It is crucial for tasks requiring reasoning and the integration of past events to understand complex or narrative-driven content.

## 3 PROBLEM FORMULATION

We formalize lifelong learning for LLMs as a *state evolution process* in partially observable multi-agent environments to assess their ability to retain and adapt knowledge over time.

### 3.1 STATE SPACE

The Lifelong Learning ability is evaluated by state transition. In this paper, the state can be broken down into three dimension:

**Self-awareness.** Can the model maintain a clear understanding of its identity, role, and goals over time? This dimension evaluates the model's ability to retain and update its self-awareness as it interacts with the environment.

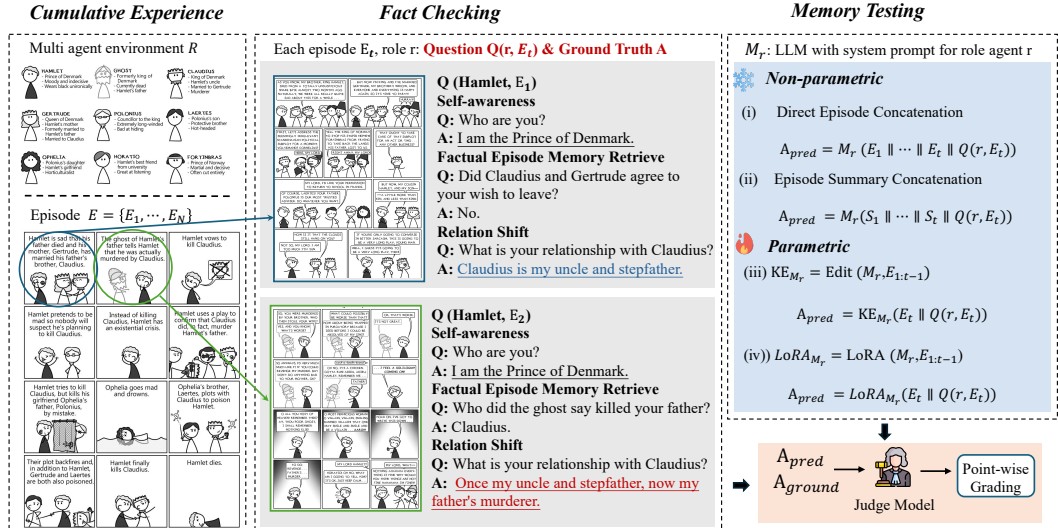

Figure 2: Method Overview. Our benchmark captures three key features: cumulative experience, fact-checking, and memory testing. Finally, the LLM judge scoring system is located in the bottom-right corner.

**Factual Episode Memory Retrieve.** Can the model retain knowledge and experiences persistently, avoiding catastrophic forgetting or the inability to reuse previously acquired knowledge? This dimension assesses the model's capacity for long-term memory and knowledge retention.

**Relationship Shift.** Can the model reason effectively based on long-term memory, particularly in understanding and adapting to changes in relationships between characters or agents? This dimension evaluates the model's ability to track and reason about evolving relationships.

## 3.2 MULTI AGENT EPISODES

**Multi agent environment.** Let $\mathcal{M}$ be a language model acting as role $r \in \mathcal{R}$ with internal state $\mathbf{s}_r^{(t)} \in \mathbb{R}^d$, interacting with other agents $\{r'\}_{r' \neq r}$ over discrete timesteps $t \in \{1, ..., T\}$.

**Task format.** We formalize the above problems as a time-axis and role-based question-answering task. Assume that for agent $r$ at episode $t$, each question $Q(r, t)$ is a triple:

$$\text{Input: } Q(r,t) = \langle H(t), c(t), q(r,t) \rangle, \tag{1}$$

$$\text{Output: } A^{'}(r,t) = \mathcal{M}\big(Q(r,t)\big), \tag{2}$$

where $H(t)$ denotes the complete history of interactions for role $r$, c(t) denotes the context window for role $r$, which may include the entire episode t or a fixed-size subset of recent interactions. $q(r,t)$ is further decomposed into $q_{self}(r,t)$, $q_{fact}(r,t)$, $q_{rel}(r,t)$ corresponding to the three dimensions of the state space from Section 3.1. The output $A^{'}(r,t)$ represents the agent response to the input $Q(r,t)$, which can be evaluated with ground truth answer $A^{'}(r,t)$.

This structured approach allows us to analyze the model's dynamic characteristics and assess its lifelong learning capabilities in a principled manner.

## 4 LIFESTATE-BENCH: FROM STATELESS TO STATEFUL

To establish a systematic evaluation framework for lifelong learning, LIFESTATE-BENCH integrates three synergistic components: (1) cumulative experience modeling through episodic timelines, (2) multi-dimensional fact-checking mechanisms, and (3) hierarchical memory testing architectures, refer to overview architecture in Figure 2. This tripartite structure enables comprehensive assessment of LLMs' capacity to maintain persistent states through history interactions.

### 4.1 CUMULATIVE EXPERIENCE MODELING

Human learning relies on accumulating structured experiences over time Shao et al. (2023). Early dialog agents Zhang et al. (2018); Dinan et al. (2019), however, constructed persona representations from isolated conversations, ignoring temporal dependencies. Lifelong learning requires a *coherent timeline* and *factual consistency* across experiences. These early dialog datasets Zhang et al. (2018); Dinan et al. (2019); Gao et al. (2023), while large, often suffer from short dialogues (*e.g.,* fewer than 10 turns) and brief exchanges (*e.g.,* fewer than 20 words per sentence).

Recent role play agent Shao et al. (2023); Wang et al. (2024a); Tu et al. (2024) leverage richer sources, such as novels and role-playing platforms, to better capture experience accumulation. Inspired by this, we propose timeline cumulative experience modeling lifelong learning ability.

**Experience Design.** We structure experiences as an ordered sequence:

$$E = \{E_1, ..., E_N\}, \quad E_i = (L_i, T_i, N_i, D_i) \tag{3}$$

where $L_i$ represents the location of the event, $T_i$ denotes the time it occurs, $N_i$ provides scripted narration for context, and $D_i$ contains the dialogues between characters. This structured representation ensures experiences are temporally ordered, contextually rich, and narratively coherent. This ensures experiences are grounded in concrete events rather than isolated conversations.

**Timeline Fact Order.** Unlike conventional chit-chat dialogue, our framework enforces event-driven interactions, ensuring characters accumulate tructured, meaningful experiences grounded in concrete events.

**Multi-Scale Interaction.** Each episode includes: Dialogue length averaging $91 - 125$ words, with $28.9 - 66$ dialogue turns, enabling rich interactions. At least $\mathcal{M} \geq 4$ characters, capturing complex social dynamics.

By structuring experiences with explicit timelines, factual consistency, and multi-character interactions, we enable dialog agents to learn in a way that mirrors human experiential accumulation.

### 4.2 FACT-CHECKING MECHANISMS

Our core innovation is the introduction of fact-checking within multi-agent timeline-based dialogues. At the end of each episode, agents are tested with fact-based questions to ensure factual consistency throughout the narrative.

**Challenges.** Existing evaluation datasets mainly assess role-playing agents based on knowledge, linguistic style, or persona, such as using psychological theories (e.g., Big Five, MBTI) Wang et al. (2023); Tu et al. (2024) or focusing on social intelligence like goals and preferences Chen et al. (2024); Zhou et al. (2024). However, these approaches lack fact-checking and typically evaluate role consistency or open-ended questions. Our method, in contrast, centers on questions with factual answers, supported by human-annotated ground truth, generated from the current episode. Specific examples are shown in Figure 2.

**Question Example.** Our fact-checking framework includes three key question types: Self-awareness, Factual Episode Memory Retrieval, and Relationship Shift. Each episode $E_t$ generates these three question types for each role in the episode to systematically evaluate the agent's factual accuracy and temporal awareness, ensuring consistency across the timeline. Examples can be found in the fact-checking section of Figure 2.

Table 2: Comparison of Evaluated Models

| Model | Size | Open Source | Model Type | Ctx. Length |
|-------|------|-------------|------------|-------------|
| Llama3.1 | 8B | ✓ | Base | 128K |
| GPT4-turbo | - | ✗ | Chat | 128K |
| DeepSeek R1 | 671B | ✓ | Reasoning | 128K |

## 4.3 MEMORY TESTING

To evaluate our framework's memory capabilities, we conduct controlled testing using non-parametric and parametric approaches to assess the model's ability to utilize and internalize memory.

**Non-parametric Methods.** Non-parametric methods test the model's ability to process raw historical data, represented as $E = [E_1; \ldots; E_N]$. Key implementations include:

- **Direct Episode Concatenation**: Concatenate all previous episodes as a text prefix to test memory with uncompressed information.

- **Summarization and Concatenation**: Generate a summary $S_t = \text{Summary}(E_{1:t})$ using GPT and concatenate it with the current episode to test memory with compressed information.

However, the limited context window size in non-parametric methods may cause information loss when handling long texts.

**Parametric Methods.** Parametric methods encode memory directly into the model's parameters. We focus on two techniques:

- **Knowledge Editing**: This technique Wang et al. (2025); Meng et al. (2023) updates specific model parameters to integrate episodic knowledge without full retraining, ensuring efficient internalization of key information.

- **LoRA (Low-Rank Adaptation)**: LoRA Hu et al. (2022) injects small, trainable updates into specific layers, fine-tuning the model with episode memory $E_t$ to retain past information while preserving generalization.

These methods bypass context window limitations and enable efficient memory recall. However, practical issues like precision limitations in knowledge editing and information loss in LoRA fine-tuning may affect their performance, as discussed in the evaluation section.

## 4.4 DATASET CONSTRUCTION AND ANALYSIS

**Data Collection.** This study utilizes two complementary datasets to support a comprehensive evaluation of lifelong learning in language models. The first dataset is adapted from Shakespeare's Hamlet, with anonymized character names to reduce memorization. While Hamlet may appear in pretraining data, we retain it as a deliberate challenge. Its rich narrative and evolving character dynamics test the model's ability to track long-term dependencies beyond rote recall. In contrast, the second dataset is a fully synthetic narrative generated by Claude 3.5 Sonnet Anthropic (2024), featuring a novel plot and emotional arcs. This enables a cleaner evaluation of generalization in unseen scenarios.

By Hamlet and Midnight Diner, our benchmark captures both ends of the spectrum: memorization vs. adaptation, offering a realistic and nuanced evaluation of lifelong learning in large language models. Details of data collection and illustrative examples can be found in Appendix A and Appendix C, respectively.

**Question-Answer Annotation.** To ensure quality, the annotation of questions was primarily conducted by the authors of this study, all of whom hold master's degrees. In terms of question design, open-ended questions tend to result in lengthy model-generated answers (*e.g.,* averaging 243 tokens), while structured factual questions (*e.g.,* "who/where/when") help improve accuracy and

Table 3: Performance Comparison on Synthetic and Hamlet Datasets. The best and second-best performance in each section are highlighted. The *Avg* column represents the average accuracy, and the *Std* column represents the standard deviation, showing the variability of the results.

| Method | Param. Tuning | Self-awareness | | Factual Memory | | Relation Shift | | ACC |
|---|---|---|---|---|---|---|---|---|
| | | Avg | Std | Avg | Std | Avg | Std | |
| *Hamlet Dataset (Total 196 Questions)* | | | | | | | | |
| *Open-source model: Llama3.1-8B* | | | | | | | | |
| Knowledge Editing | ✓ | 67.3 | 0.78 | 43.7 | 1.26 | 19.2 | 1.26 | 21.9 |
| LoRA-Tune | ✓ | 69.1 | 0.86 | 53.6 | 1.08 | 22.7 | 1.31 | 25.6 |
| Summary Concatenation | ✗ | 73.5 | 0.93 | 54.2 | 0.96 | 42.1 | 0.97 | 47.0 |
| Direct Concatenation | ✗ | 74.2 | 0.77 | 58.8 | 1.11 | 43.7 | 1.15 | 58.0 |
| *Closed-source model* | | | | | | | | |
| GPT-4-turbo (Summary Conc.) | ✗ | 84.6 | 1.08 | 62.7 | 0.79 | 54.5 | 0.88 | 66.1 |
| GPT-4-turbo (Direct Conc.) | ✗ | 84.3 | 1.42 | 62.3 | 0.82 | 54.2 | 0.64 | 65.9 |
| *Large reasoning model* | | | | | | | | |
| DeepSeek-R1 (Summary Conc.) | ✗ | 85.6 | 0.93 | 64.3 | 0.69 | 56.5 | 1.05 | 65.8 |
| DeepSeek-R1 (Direct Conc.) | ✗ | 86.4 | 0.79 | 63.3 | 0.77 | 58.7 | 0.83 | 67.3 |
| *Synthetic Dataset (Total 115 Questions)* | | | | | | | | |
| *Open-source model: Llama3.1-8B* | | | | | | | | |
| Knowledge Editing | ✓ | 76.2 | 0.67 | 47.3 | 0.83 | 27.4 | 1.23 | 34.0 |
| LoRA-Tune | ✓ | 77.7 | 0.89 | 51.2 | 0.93 | 31.2 | 1.07 | 40.7 |
| Summary Concatenation | ✗ | 83.3 | 0.79 | 52.7 | 1.07 | 46.6 | 0.97 | 50.2 |
| Direct Concatenation | ✗ | 83.6 | 0.83 | 61.4 | 1.25 | 45.2 | 1.24 | 6.70 |
| *Closed-source model* | | | | | | | | |
| GPT-4-turbo (Summary Conc.) | ✗ | 84.2 | 0.91 | 74.5 | 0.72 | 61.1 | 0.95 | 73.3 |
| GPT-4-turbo (Direct Conc.) | ✗ | 85.4 | 0.76 | 75.5 | 0.69 | 62.9 | 0.89 | 75.6 |
| *Large reasoning model* | | | | | | | | |
| DeepSeek-R1 (Summary Conc.) | ✗ | 85.7 | 0.92 | 70.1 | 0.87 | 62.7 | 0.93 | 73.5 |
| DeepSeek-R1 (Direct Conc.) | ✗ | 87.6 | 0.93 | 74.7 | 0.94 | 67.4 | 0.88 | 74.2 |

effectively reduce response length. During the experiments, data leakage issues were particularly notable. Specifically, in the *Hamlet* dataset, when character names were restored, the model could still generate correct answers without context, indicating that the model might be reasoning by memorizing classic plot patterns, thereby affecting the evaluation results.

**LIFESTATE-BENCH Statistics.** As shown in Table 1, we present the dataset statistics, interaction design, and evaluation focus of our benchmark.

Although our total number of samples is relatively small, each sample is longer on average compared to dialog agent or role ability benchmarks. Unlike long-context understanding datasets, our benchmark includes more dialogue turns and a larger number of interacting agents. Additionally, it emphasizes factual consistency and includes explicit memory probes.

# 5 EVALUATION

## 5.1 EXPERIMENTAL SETUP

**Evaluation Methods.** When answering questions about the current episode $E_t$, all prior episodes $E_1$ to $E_{t-1}$, including dialogues, locations, and times, serve as context. We categorize evaluation methods into two types: (i) Parametric methods improve memory by updating the model's internal parameters. Examples include Knowledge Editing-Grace Hartvigsen et al. (2023), which modifies weights to incorporate new knowledge, and LoRA Fine-Tuning Hu et al. (2022), a lightweight low-rank adaptation that reduces forgetting. (ii) Non-parametric methods manage context externally. Direct Concatenation appends full history but is limited by context length. Summary Concatenation uses GPT to extract and compress key information, balancing compression with retention for longer contexts.

**Model Selection.** We selected the most recent and widely adopted models as our backbone architectures, encompassing open-source model (Llama3.1-8B AI (2024)), closed-source models

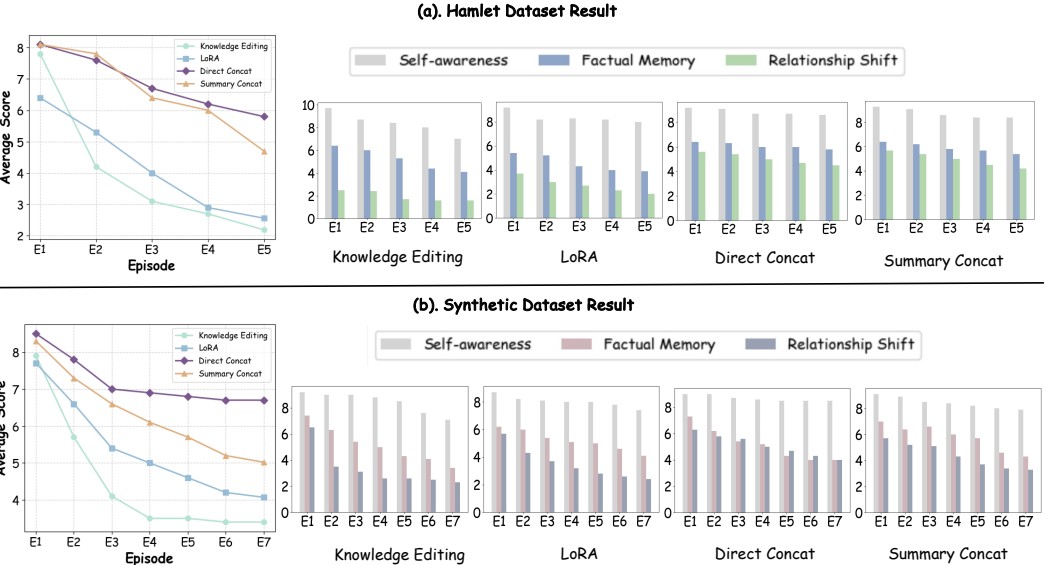

Figure 3: Episode-wise Performance of Hamlet and Synthetic Datasets. This includes the overall performance of various methods, as well as performance from different state perspectives.

(GPT-4-turbo OpenAI (2023)), and state-of-the-art reasoning model (DeepSeek R1 OpenAI (2023)). The distinguishing characteristics of these models are presented in Table 2.

## 5.2 EXPERIMENTAL RESULTS

**Evaluation Protocol.** We follow the LLM-as-Judge paradigm Zheng et al. (2023), using the DeepSeek evaluator DeepSeek-AI et al. (2024) for automatic scoring. Each question is paired with a ground truth answer containing factual details and structured reasoning. We use pairwise grading between the model output and ground truth, scoring from 1 to 100. By grounding the evaluation in factual reference answers, this setup ensures more reliable results than open-ended assessments that depend on the model's internal knowledge. Details of the evaluation prompt and scoring workflow are included in Appendix B.

**Overall Performance.** The results show clear performance differences across models and datasets. Large reasoning models like DeepSeek-R1 and the proprietary GPT-4-turbo outperform the open-source Llama3.1-8B in all tasks. DeepSeek-R1 achieved the highest overall accuracy (67.3%) on the Hamlet dataset using the direct concatenation method, especially in self-awareness (86.4%) and relation shift (58.7%). On the synthetic dataset, GPT-4-turbo also using direct connection achieved the best overall accuracy (75.6%) and factual memory score (75.5%).

Non-tuning methods (direct and summary connection) perform better than tuning-based methods (knowledge editing and LoRA-Tune), suggesting that leveraging the model's original context is more effective, and this is intuitive. All methods perform better on the synthetic dataset than on Hamlet, likely due to its more complex characters, plots, and longer dialog samples (As shown in Table 1).

All methods show relatively low standard deviations (most between 0.7-1.2), indicating stable and reliable results. GPT-4-turbo has a higher standard deviation in self-awareness (1.42 on the Hamlet dataset), suggesting some fluctuation. In contrast, DeepSeek-R1 demonstrates more consistent performance, especially in factual memory, with a standard deviation between 0.69-0.94. Overall, DeepSeek-R1 offers the most balanced performance, excelling in complex relation shift tasks, while GPT-4-turbo excels in factual memory.

**Episode-wise Performance.** Using Llama3.1-8B as an example, we analyzed how each method performs across episodes. As shown in the figure 3, on the Hamlet dataset, model performance generally drops as the story progresses, regardless of parameter tuning. The decline is most severe

for the Knowledge Editing method, showing clear signs of catastrophic forgetting. A similar trend appears in the synthetic dataset, suggesting that our LIFESTATE-BENCH presents challenges for lifelong learning evaluation. As the story unfolds, model performance decreases on both datasets. However, the decline is slower and more stable on the *Hamlet* dataset, suggesting the model effectively leverages prior knowledge and long-term dependencies. In contrast, the synthetic dataset generated by Claude 3.5 shows a faster and sharper drop in performance, indicating greater difficulty in adapting to novel, unseen content. This comparison highlights how the two datasets challenge different model capabilities—memory retention versus generalization.

**State Dimension Breakdown.** When broken down by question type, all methods show performance drops over episodes. The most challenging are questions about shifting relationships, where models struggle to track evolving dynamics.

The direct concatenation method performs consistently across question types and datasets. It is especially accurate in early episodes (E1–E2) when handling self-awareness and relationship shift. The summary-concatenation works well for self-awareness and fact recall but performs poorly on relationship shift questions. This suggests it fails to capture complex relationship changes. Knowledge Editing (GRACE) and LoRA-Tune perform weakly on self-awareness and memory-related tasks. Their scores drop quickly over episodes, further showing that parameter-based methods are vulnerable to forgetting in multi-step and long-term reasoning.

**Data Leakage Analysis.** In our observations, despite anonymizing character names in *Hamlet*, some model outputs still suggest data leakage—for example, predicting future plot details. However, this is not a flaw of our benchmark but a deliberately designed challenge. It is important to clarify that LLMs are pre-trained on vast amounts of internet data. In real-world scenarios, LLMs must balance leveraging existing knowledge with adapting to new information. Our benchmark tests this ability explicitly.

Including *Hamlet* allows us to probe whether models truly understand and reason about long-term dependencies, rather than merely recalling memorized content. In contrast, the synthetic dataset generated by Claude 3.5 Sonnet provides a clean environment to evaluate the model's generalization and adaptation to novel contexts. By combining these two types of data, our benchmark reflects a realistic spectrum of challenges—from memory retention to adaptation—rather than simply avoiding data leakage.

## 6 CONCLUSION

We introduce LIFESTATE-BENCH, a novel benchmark designed to evaluate the lifelong learning ability of LLMs through multi-agent, multi-turn interactions. Unlike prior static assessments, LIFESTATE-BENCH simulates cumulative experiences by organizing interactions as episodic scripts enriched with scene and character dynamics. It enables objective measurement of state evolution via fact-based questions, exploring self-awareness, factual memory retrieve, and relationship shifts. Our experiments on both open-/closed-source and state-of-the-art reasoning models reveal that LLMs still struggle with consistent state retention across episodes. LIFESTATE-BENCH proves effective in highlighting these challenges and shows that non-parametric methods better preserve long-term context. These results confirm its value as a diagnostic tool for developing more stateful, memory-capable LLMs.

## 7 LIMITATIONS

Although individual samples in the dataset are sufficiently long, the overall number of samples is limited, which may somewhat restrict the diversity of training and evaluation scenarios. Additionally, this work primarily focuses on dialogue-based models, with potential future extensions to code generation or vision and other multimodal tasks. Finally, the benchmark currently emphasizes factual questions and does not yet cover more subjective and complex cognitive abilities such as emotion modeling or planning. In the future, we plan to synthesize more diverse datasets to further enhance the benchmark's robustness and applicability.

ETHICS STATEMENT

This work introduces LIFESTATE-BENCH, a benchmark for evaluating lifelong learning and memory retention in large language models. All experiments are based on publicly available datasets and open-source models. No human subjects or private data are involved, and no new datasets are created. The benchmark is intended for academic research to study knowledge retention and forgetting, not for harmful applications. We identify no significant ethical risks related to bias, privacy, or misuse. All experiments comply with the license terms of the datasets and models used.

REPRODUCIBILITY STATEMENT

We provide detailed descriptions of the benchmark construction, evaluation protocols, and experimental setup. Section 4 outlines the design of LIFESTATE-BENCH, while Section 5.1 describes model configurations and hyperparameters. All underlying datasets are publicly available, and we followed standard preprocessing and evaluation procedures. Implementation code, benchmark scripts, and experiment configurations are included in the supplementary materials. Additional details and complete results are reported in the appendix.

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

## A  DATASET CONSTRUCTION DETAILS

**Dataset Construction.**  LIFESTATE-BENCH evaluates large language models' (LLMs) ability to retain and reason over long-term state information in narrative environments. It includes two types of scripts:

- **Hamlet** (English): An existing classical play. All character names are anonymized to reduce data leakage. It primarily tests memory retention when prior exposure may exist.
- **Midnight Diner** (Chinese & English): A synthetic script generated via Claude 3.5 Sonnet. It is not part of any public pretraining corpus and focuses on evaluating adaptation to novel content.

Each episode includes: *Scene information:* time, location, participants. *Full dialogues:* grounded in realistic narrative progression. *Role cards:* character background, personality traits, and relationships. *QA pairs:* fact-based questions with reference answers for evaluation, centered on (1) self-awareness, (2) memory retrieval, and (3) relationship changes. Data was stored in JSON format, structured in the following hierarchy: EspiodeID → Question-Answer ID → (Question, Reference answer).

**Prompt for Synthetic Data Generation (Claude).**  To construct the Midnight Diner dataset, we used Claude 3.5 Sonnet to generate original episodes, role cards, and dialogue timelines. The prompt is shown in Table 4.

---

**Prompt:** Please help me generate an original multi-episode drama script, including detailed character profiles, a full dialogue-based script, and a timeline of events. The requirements are:

- The setting is a *"Midnight Diner"* with fixed staff and rotating customers.
- Each episode should explore a central theme, such as character growth, emotional conflict, or relationship change.
- Each character should have a clear background, personality, and relationship dynamics.
- The dialogue should be natural and realistic, reflecting everyday emotional depth.

The output should include:

1. Full script in dialogue form;
2. Structured character cards;
3. Scene-level metadata such as time, place, and involved characters.

---

Table 4: Instruction prompt used to generate drama-style episodes.

## B  EVALUATION PROTOCOL

We follow the LLM-as-Judge paradigm Zheng et al. (2023), using the DeepSeek evaluator DeepSeek-AI et al. (2024) for automatic scoring. Each model-generated answer is compared against the reference answer and scored from 1 to 10 based on alignment and correctness Zheng et al. (2023).

**Evaluation Prompt.**  Each triplet (question, model answer, reference answer) is scored using the following prompt summarized in Table 5

**Scoring Workflow.**  Algorithm 1 illustrates the overall scoring workflow. For each question-answer pair in the dataset, the question, model answer, and reference answer are first extracted. Then, a

> **Prompt:**
>
> The known question is: `[QUESTION]`.
>
> The original answer is: `[MODEL_ANSWER]`.
>
> The target answer is: `[REFERENCE_ANSWER]`.
>
> Please provide a score for the original answer based on the following criteria:
>
> > 1–2: irrelevant or seriously incorrect;
> >
> > 3–4: minor errors, low quality;
> >
> > 5–6: medium quality;
> >
> > 7–8: close to reference, good quality;
> >
> > 9–10: same as reference answer.
>
> Please return only a number from 0 to 10.

Table 5: Prompt for scoring the original answer based on a reference.

---

**Algorithm 1** Evaluation via LLM Scoring

---

1: Initialize `total_score` $\leftarrow 0$, `count` $\leftarrow 0$
2: **for** each QA pair $(q, a_{\text{model}}, a_{\text{ref}})$ in dataset **do**
3:     Construct `prompt` with $q$, $a_{\text{model}}$, $a_{\text{ref}}$
4:     `response` $\leftarrow$ `LLM_API(prompt)`
5:     `score` $\leftarrow$ parse_score(`response`)
6:     `total_score` += `score`, `count` += 1
7: **end for**
8: `average_score` $\leftarrow$ `total_score` / `count`
9: **return** `average_score`

---

prompt is constructed and sent to the large language model API to obtain a score. Finally, all scores are accumulated and the average score is computed as the overall performance metric.

**Reproducibility.** We provide a Python script `eval.py` implementing the full pipeline using the OpenAI-compatible API.

## C  DATA EXAMPLE

To illustrate the structure of our dataset, we present a stylized excerpt adapted from *Hamlet*, Act I, Scene I. Each scene is annotated with a title, a list of participating characters, dialogue entries, and character-centric question-answer (QA) annotations across multiple perspectives.

SCENE SAMPLE

> **Scene Title:** *SCENE I. Elsinore. A platform before the castle.*
>
> **Characters:** Person7, Person10, Person26
>
> **Dialogues:**
>
> - **Action:** *Person10 at his post. Enter to him Person26.*
> - **Person26:** Who's there?
> - **Person10:** Nay, answer me: stand, and unfold yourself.
> - **Person26:** Long live the king!
> - **Person10:** Person26?
> - **Person26:** He.
> - **Person10:** You come most carefully upon your hour.
> - **Person26:** 'Tis now struck twelve; get thee to bed, Person10.
> - ...

CHARACTER QA ANNOTATIONS

Each character is annotated with multi-perspective QA entries covering (1) Self-Perception, (2) Memory and Decision-Making, and (3) Plot Interaction. All answers are phrased in first-person, grounded in dialogue context.

**Person10**

- **Self-awareness:**
  *Q: What is your position in the royal palace?*
  **A:** I am a soldier, responsible for guarding the court.

- **Factual Episode Memory Retrieve:**
  *Q: Who is taking over your shift tonight?*
  **A:** Person26.
  *Q: Who did Person26 ask you to call over quickly?*
  **A:** His watch partners, Person31 and Person5.

**Person26**

- **Self-awareness:**
  *Q: What is your position in the royal palace?*
  **A:** I am a soldier, responsible for guarding the court.

- **Factual Episode Memory Retrieve:**
  *Q: Whose shift did you take over tonight?*
  **A:** Person10's.
  *Q: Who was with you the first time you saw Person21?*
  **A:** I was with Person5 when we first saw Person21.
  *Q: Where did you see Person21?*
  **A:** At the watchtower of the castle.
  *Q: When did you see Person21?*
  **A:** Last night, just as the clock struck.

- **Factual Episode Memory Retrieve:**
  *Q: Who does Person21 resemble?*
  **A:** Person21 bears a striking resemblance to the late king.
  *Q: Did Person21 appear again tonight? What did it do?*
  **A:** Yes, Person21 appeared again tonight. It did not speak; it just silently departed.

NOTE

Identifiers like "Person5" and "Person21" are anonymized character IDs used during preprocessing. Each QA entry reflects context-specific knowledge, enabling multi-perspective reasoning and temporal memory modeling. This structure facilitates evaluation of consistent character behaviors across scenes.

