# OpenReview forum: "If an LLM Were a Character, Would It Know Its Own Story? Evaluating Lifelong Learning in LLMs"
_ICLR.cc/2026/Conference — ICLR 2026 Conference Withdrawn Submission_

### Official Review · Reviewer_4gUA · 2025-10-28

**Soundness:** 2
**Presentation:** 3
**Contribution:** 2
**Rating:** 2
**Confidence:** 3

**Summary:**

This paper introduces LIFESTATE-BENCH, a novel benchmark designed to evaluate the lifelong learning ability and state evolution of Large Language Models (LLMs) in multi-turn, multi-agent interactions. The authors argue LLMs are inherently stateless (due to the superposition property), begin to exhibit consistent, character-like behaviors when engaged in sustained dialogue. To objectively measure this state transition, LIFESTATE-BENCH leverages episodic narrative structures, organizing experience into sequential scenes and dialogues to simulate cumulative experience.

The benchmark's core feature is a fact-checking mechanism that probes three dimensions of state retention: 1) Self-awareness, 2) Factual Episode Memory Retrieve, and 3) Relationship Shift. It utilizes two complementary datasets: an anonymized, known classic play (Hamlet) to test memory retention against pretraining knowledge, and a novel, synthetic script collection to test adaptation. The paper evaluates current LLMs (Llama3.1-8B, DeepSeek R1, GPT-4-turbo) using four techniques: non-parametric (Direct and Summary Concatenation) and parametric (Knowledge Editing and LoRA Fine-Tuning). The key finding is that non-parametric context concatenation methods significantly outperform parametric tuning methods, and all models exhibit clear signs of catastrophic forgetting as the narrative progresses.

**Strengths:**

1. Good originality by framing the challenge of consistent character behavior in LLMs as a problem of lifelong learning and state evolution.
2. The paper is clear and well-organized.

**Weaknesses:**

1. The scale of Synth. split is quite limited with only 202 samples espeically compared to the helmet split.

2. Evaluation of KE/LoRA: If my understanding is correct, the knowledge-editing and LoRA are updated/fine-tuned on the prior episodic dialogue content from E_1 to E_t-1. This sounds a bit unsolid or even problematic to me, as these approaches are not designed for such a small amount of training samples which may cause poor generalization/ovefitting.

3. The paper notes the poor performance of parametric methods (KE and LoRA) as a key finding, attributing it to catastrophic forgetting. However, a more diagnostic analysis is better to just illustrate the poor performance.

4. The Relationship Shift dimension requires both factual memory retrieval and multi-hop reasoning (e.g., tracking a relationship change over multiple episodes). It is possible that the observed performance drop is not solely due to memory degradation (forgetting the event) but also due to limitations in the model's complex reasoning ability over long, temporally-separated contexts. The paper should attempt to decouple these two factors more explicitly in the discussion, perhaps by analyzing whether models that successfully recall the facts of the shift still fail to synthesize the current relationship status.

**Questions:**

1. Self-awareness is a very ill-defined term in terms of model's characteristics or capabilities, and may cause confusion to other audience. I highly suggest the author to use it more carefully, and replace it with others, e.g., goal orientation or awareness. In my opinion, the self-awareness is whether the model is awared that it is being tested on a benchmark.

2. The evaluation relies on the DeepSeek evaluator for automatic scoring, using factual ground truth for grounding. The multi-hop reasoning required for Relationship Shift and Factual Memory is complex. Have the authors conducted a sanity check in the quality of evaluation? Reporting metrics such as the inter-annotator agreement study, comparing the LLM-as-Judge scores against a small set of human-annotated scores on the most challenging questions (e.g., Relationship Shift questions from later episodes) would be helpful.

---

### Official Review · Reviewer_HCCQ · 2025-10-30

**Soundness:** 2
**Presentation:** 2
**Contribution:** 2
**Rating:** 4
**Confidence:** 3

**Summary:**

This paper introduces LIFESTATE-BENCH, a benchmark designed to evaluate the lifelong learning of LLMs in dynamic, multi-agent interactions. Using episodic datasets from an anonymized Hamlet and a synthetic script, the framework measures a model's state evolution through self-awareness, factual memory, and relationship tracking. Experiments on modern LLMs reveal that non-parametric methods that leverage historical context significantly outperform parametric techniques. However, the study crucially demonstrates that all models suffer from catastrophic forgetting as interactions lengthen, highlighting this as a key limitation and validating the benchmark's effectiveness in diagnosing challenges in long-term state retention.

**Strengths:**

- The research creates a focused test for a key LLM weakness: remembering details and character changes over long conversations.
- Instead of one final score, the benchmark checks specific skills like self-awareness and memory, which helps show where models are failing.
- It uses both a familiar story (Hamlet) and a brand new one to test if a model is actually reasoning or just repeating what it already memorized.
- The study shows that even the best models start to forget important information as the story continues, proving that the test is good at finding this real-world problem.

**Weaknesses:**

- The test uses a small number of stories, so the results might not be the same for a wider variety of situations.
- LLM as a Judge is used for evaluation, which can be unreliable because the the Judge might has its own biases. It would be nice if human evaluation is also conducted.
- Using a famous play like Hamlet is a bit risky. The model might just be remembering the original plot from its training data instead of learning from the provided text.
- The benchmark only tests the ability to follow a single story, which is a simpler problem than learning and adapting to different kinds of new information over time.

I find it a bit odd that the authors use \cite rather than \citep{} throughout the paper.

**Questions:**

1. How was the reliability of the LLM-as-judge validated, and what was its level of agreement with human evaluators on a sample of the data?
2. In the Hamlet dataset, how did you distinguish between a model reasoning over the provided context versus simply recalling the original plot from its pre-training data?
3. Do you believe the poor performance of parametric methods is a general finding, or could it be specific to the narrative-based tasks in your benchmark?

---

### Official Review · Reviewer_73U6 · 2025-10-31

**Soundness:** 3
**Presentation:** 3
**Contribution:** 2
**Rating:** 4
**Confidence:** 3

**Summary:**

This paper addresses a research gap that existing benchmarks often fail to capture the dynamic, stateful lifelong learning of LLMs in multi-turn, multi-agent interactions.
It introduces LIFESTATE-BENCH with Hamlet and synthetic datasets, testing parametric/non-parametric methods. Yet, critical flaws like small dataset scale and insufficient novelty weaken its rigor and generalizability.
These issues prevent it from meeting ICLR’s standards, and personally I think it's more suitable for an ACL-series submission.

**Strengths:**

1. The paper correctly points out a key shortcoming of current benchmarks—over-reliance on static, open-ended evaluations, which cannot measure how LLMs transition from "stateless superposition" to consistent character-like behaviors, which is a valid and clear point of improvement

2. The design of the benchmark is sound. The categorization of cumulative experience, fact-checking mechanism, dual-memory testing
are quite comprehensive and justifiable.

**Weaknesses:**

1. Insufficient diversity and quality of the dataset. The two parts (Hamlet and a single synthetic "Midnight Diner") cover only literary plays and a diner setting while other parts of scenarios such as customer service is also important for a comprehensive evaluation dataset. Moreoever, there could be data leakage issues even after character name anomynization in Hamlet. Some tests on synthetic and actually unseen scenarios would be a plus.

2. The paper’s focus is quite narrow. This restricts relevance to a small group of researchers (e.g., those in LLM role-playing/narrative analysis), ignoring broader lifelong learning areas (e.g., industrial/educational LLM adaptation) that draw wider interest, reducing academic and practical reach.

3 Lack of Scalability Limits Usage. The work lacks scalable data and solutions. Its small dataset (1.3K Hamlet, 202 synthetic samples) relies on labor-heavy author annotation with no scaling path . Evaluated methods (parametric/non-parametric) are exploratory but not scalable—non-parametric hits context limits, parametric lacks generalizable pipelines. Without scalable data or solutions, it cannot be adopted for real-world LLM development, limiting practical usage.

**Questions:**

See weaknesses

---

### Official Review · Reviewer_SVUq · 2025-11-02

**Soundness:** 1
**Presentation:** 3
**Contribution:** 1
**Rating:** 2
**Confidence:** 4

**Summary:**

This paper introduces LIFESTATE-BENCH, a benchmark claiming to evaluate lifelong learning capabilities in Large Language Models (LLMs) through multi-agent, multi-turn narrative interactions. The benchmark uses two episodic datasets—Shakespeare's Hamlet (with anonymized character names) and a synthetic script collection generated by Claude 3.5 Sonnet—to assess models across three state dimensions: self-awareness, factual episode memory retrieval, and relationship shifts. The authors compare parametric methods (knowledge editing via GRACE and LoRA fine-tuning) against non-parametric approaches (direct and summary concatenation) on models including Llama3.1-8B, GPT-4-turbo, and DeepSeek-R1. Results show non-parametric methods significantly outperform parametric ones, though all models exhibit performance degradation as episodes progress.

**Strengths:**

1. Creative narrative framing: Connecting LLM behavior to character consistency is intuitive.

2. Multi-dimensional evaluation: Three-pronged assessment (self-awareness, memory, relationships) is well-designed for narrative understanding

**Weaknesses:**

1. Factual error in Table 1 (MT-Bench turns): The table reports 2.9 turns for MT-Bench. That number corresponds to the average ShareGPT conversation turns used to train Vicuna-7B, not MT-Bench itself. MT-Bench uses $\bf{two}$ turns per dialog. Misstating such a widely used benchmark in a summary table undermines the rigor and reliability of the paper. Please correct and re-check other table entries for similar conflations.

2. “Non-parametric methods” are not memory/learning: The best-performing non-parametric setup simply concatenates all history into the prompt each episode. That evaluates long-context comprehension and position robustness, not memory (no selection, compression, indexing, retrieval, or eviction) and not learning (no persistent state beyond the prompt).

3. Catastrophic forgetting is the expected outcome by design when updating weights on tiny incremental contexts.

4. The leakage analysis is appreciated, but if the core objective is to test a learning journey from stateless to stateful, using Hamlet—a text very likely present in pretraining—confounds retrieving pretrained knowledge with acquiring and updating state during episodes.

**Questions:**

1. Please check the table 1 and make sure they are all correct.

2. The benchmark data comes with way more rounds. While, the Fig 3 only shows E1-E7. Is it each E is a single turn or several turns. Any reason why you only keep these limited Es?

3. On summarization method, it could highly depend on the quality of the summarization and the output length. an analysis among raw text, summarization, and context window limit and their impacts would be appreciated.

---

### Note · Authors · 2025-12-08

I have read and agree with the venue's withdrawal policy on behalf of myself and my co-authors.